# Towards Robust Visual Tracking for Unmanned Aerial Vehicle with Spatial Attention Aberration Repressed Correlation Filters

Zhao Zhang †, Yongxiang He †, Hongwu Guo *, Jiaxing He, Lin Yan and Xuanying Li

College of Intelligent Science and Technology, National University of Defense Technology, Changsha 410000, China; zhangzhao981124@163.com (Z.Z.); heyongxiang1995@163.com (Y.H.); hejiaxing211@163.com (J.H.); yeln.w@outlook.com (L.Y.)
* Correspondence: guohongwu@nudt.edu.cn
† These authors contributed equally to this work.

**Abstract:** In recent years, correlation filtering has been widely used in the field of UAV target tracking for its high efficiency and good robustness, even on a common CPU. However, the existing correlation filter-based tracking methods still have major problems when dealing with challenges such as fast moving targets, camera shake, and partial occlusion in UAV scenarios. Furthermore, the lack of reasonable attention mechanism for distortion information as well as background information prevents the limited computational resources from being used for the part of the object most severely affected by interference. In this paper, we propose the spatial attention aberration repressed correlation filter, which models the aberrations, makes full use of the spatial information of aberrations and assigns different attentions to them, and can better cope with these challenges. In addition, we propose a mechanism for the intermittent learning of the global context to balance the efficient use of limited computational resources and cope with various complex scenarios. We also tested the mechanism on challenging UAV benchmarks such as UAVDT and Visdrone2018, and the experiments show that SAARCF has better performance than state-of-the-art trackers.

**Keywords:** unmanned aerial vehicle; correlation filtering; visual tracking; aberration suppression; spatial attention; global context





## 1. Introduction

In recent years, UAVs have been more and more widely used in many fields, such as civil, military, and scientific research, by virtue of their small size, flexible movements, and easy control, for example, in power line detection in harsh environments, atmospheric environment detection, rescue and disaster relief, enemy reconnaissance, enemy target tracking, and search for battlefield intelligence [1–6]. UAV visual target tracking is one of the fundamental and challenging tasks with high research value, and it is also a major research hotspot in the field of computer vision at present. However, compared with general tracking scenarios, UAV visual target tracking faces more complicated challenges. To sum up, there are mainly the following points: (1) when UAVs perform tracking tasks, interference from wind and other factors leads to motion blur, camera shake, and frequent viewpoint changes, which in turn leads to the tracking drift and target loss. (2) Due to the open aerial view and complex ground environment, many interfering objects, mutual interference between targets and targets and between targets and backgrounds, poor distinguishability leads to the target model's variability and exclusivity being not so high, and it is difficult to establish an accurate target model. (3) When the UAV flies at a certain altitude, the image shadow width becomes larger, the resolution and clarity become lower, the scale of the target to be tracked on the ground becomes small, and the target features and textures become sparse, making it difficult to extract the target features and the feature representation is not significant, leading to greater difficulty in tracking [7]. Therefore, a robust UAV target tracking method is needed to cope with various complex tracking scenarios.

Deep learning has achieved better results in vision tracking in recent years, but it is still difficult to deploy deep learning tracking algorithms in UAVs for a wide range of applications in real flight scenarios due to limited on-board computational resources, small battery capacity, low power consumption, and maximum load limitations [8]. Under the hard constraints of real-time vision-based UAS, the ideal tracker should have higher computational power for sensor fusion, advanced control, etc. [9]. Although the discriminative correlation filter (DCF)-based approach cannot compete with the deep learning-based approach in terms of accuracy and precision, it can quickly learn the model from a single frame without the need for additional dataset training. Due to the use of fast Fourier transform (FFT) to transfer the evaluation of correlation to the frequency domain, it has high computational efficiency and excellent robustness on a single CPU. This method has been widely applied in the industry.

DCF uses cyclic shifting on a single image frame to generate a large number of samples for training, but due to the limited search area, it is difficult to avoid generating boundary effects and degrading the tracking performance [10]. The background-aware filter (BACF) directly constructs a cropping matrix by circularly shifting in the whole image picture to obtain more negative samples from the background, which effectively mitigates the boundary effect. Due to the expanded search area, it is able to track targets with relatively high velocities with the camera or UAV. However, as more background information is introduced, many background clutters become byproducts. This will lead to similar objects around the target as well as the background being easily identified as the original target we want to track. Huang et al. introduced an aberration suppression term based on BACF, which provides a good suppression of the aforementioned aberrations. There are also works that consider learning context information during training to extend the acceptance domain of DCF and improve the filter's ability to discriminate the background [11,12]. These methods have more limited limitations for target drift caused by fast target movement, camera shake, and partial occlusion, and lack reasonable attention mechanisms for aberration information as well as background information to devote limited computational resources to the part of the object most affected by the interference.

The attention mechanism (AM) is a data-processing method in machine learning, which is widely used in various different types of machine learning tasks such as natural language processing (NLP), image processing (CV) and speech recognition [13]. The attention mechanism is able to select and focus on relevant information in a targeted manner according to the input context and task requirements, which improves the accuracy and efficiency of the model. Inspired by spatial attention, in this paper, we propose an aberration-repressed correlation filter (SAARCF) based on the spatial attention framework to more efficiently cope with the target tracking problem in various complex scenes from the UAV perspective under the condition of limited computational resources.

First, we introduce an aberration model built from two consecutive response frames, which assigns equal attention to aberrations near the target and aberrations occurring far away from the target; however, aberrations occurring far away from the target are often irrelevant, and aberrations occurring closer to the target have a greater impact on the filter training and are more likely to cause target drift. To address this, we designed an attention module to optimize the filter's perception of aberrations by assigning different levels of attention to them according to their spatial information, thus improving the filter's ability to discriminate between targets. To further optimize the use of computational resources, we used the APCE to determine whether to learn the global context in the aberration frame. For the aberration frame, a low APCE indicates that the tracker can no longer cope with the drastic changes in the scene by relying on its own robustness, and the global context needs to be learned to introduce more background information to cope with the drastic changes in the scene.

The main contributions of this work are listed as follows:

1. We propose a robust target tracking method (SAARCF), which alleviates boundary effects and fully utilizes the spatial information of aberrations to suppress sudden changes in the response map, and can better cope with tracking problems in complex environments from the perspective of unmanned aerial vehicles.
2. A mechanism of intermittent learning global context is proposed to utilize the background information more efficiently and cope with the drastic changes in the scene under the limited computational resources.
3. SAARCF is tested on challenging UAV datasets such as UAVDT, Visdrone2018, and the experimental results show that SAARCF has a better tracking performance.

The rest of this paper is organized as follows: Section 2 provides a brief review of the prior work related to this work. Section 3 presents the baseline tracker on which our tracker is based and elaborates our proposed method. Section 4 shows the experimental results of our proposed method tested on UAV benchmarks such as UAVDT and Vistrone2018. Section 5 provides the conclusion.

## 2. Related Works

Visual object tracking methods are mainly divided into two categories: discriminative methods [14–16] and generative methods [17–19]. Generative tracking methods essentially learn a feature template in the target area of the first frame and search for the area with the highest similarity in the search area of subsequent frames. However, this approach does not model the target's appearance, treats it as a point mass, and mostly ignores background information, making it difficult to cope with complex tracking scenarios. Discriminative tracking methods, on the other hand, essentially extract features near the target and train a classifier that can distinguish between the target and the background. Discriminative tracking methods include those based on correlation filters and those based on deep learning. Discriminative methods solve the problem of insufficient samples and can extract more useful information, thus better dealing with various complex tracking environments and gradually becoming the mainstream method in the tracking field. This section mainly reviews discriminative tracking methods based on correlation filters.

### 2.1. Discriminative Correlation Filtering Algorithm

The correlation filter-based target tracking method was introduced by Bolme et al. [20] in 2010. They proposed the minimum output sum of the squared error (MOSSE) tracking algorithm. The core of this algorithm is to train a filter to perform correlation operations with training samples and minimize the squared error between the correlation output and the expected output. At the same time, the paper uses fast Fourier transform to transfer the convolution operation to the frequency domain, updating the target position through the response map. MOSSE has a high speed of up to 615 frames/s, but it does not have an advantage in tracking accuracy.

Henriques proposed the circulant structure of tracking-by-detection with kernels (CSK) algorithm based on MOSSE, which introduces the circulant matrix. Unlike previous tracking methods that used affine transformations to obtain sparse samples, CSK obtains a large number of dense samples through cyclic shifts. Due to the improvement in sample quality and quantity, CSK achieved a significant increase in tracking accuracy compared to MOSSE. CSK only uses grayscale features, so it has poor robustness in the complex scenarios of drone scenes. Therefore, Dalal and others introduced histograms of oriented gradients (HOG) features based on CSK and built the KCF algorithm [21]. Danelijian and others introduced commonly used color attribute features (CN) in target recognition and detection to improve the tracking performance based on CSK [22].

Bertinetto designed the staple tracker [23] to address the challenges of the target deformation and illumination changes. The staple tracker contains two correlation filters, using HOG and CN features, respectively. After obtaining their respective response maps, they are fused to achieve target tracking.

*2.2. Previous Work on Suppressing Boundary Effects*

The introduction of circulant matrices in the CSK algorithm leads to negative boundary effects. Boundary effects mainly refer to the generation of some inaccurate negative samples in the DCF method due to the use of the periodicity assumption of samples, which can reduce the discriminative ability of the model during the training process [24]. Generally, filters use cosine windows to mitigate boundary effects, but the effect is minimal.

In view of this, the spatially regularized discriminant correlation filter (SRDCF) [24] introduces spatial penalty weights during the filter training process to penalize the background. However, the added regularization constraint breaks the closed-form solution of ridge regression, so the optimal correlation filter parameters can only be obtained through the computationally complex Gauss–Seidel method. As a result, the algorithm's robustness is significantly improved, but the processing speed is reduced to 5 frames/s [24]. Many other studies have also focused on adding different constraints during the filter training process to obtain more discriminative filters. For example, CFLB [25] increases the proportion of effective negative samples by enlarging the training sample size and learning smaller-scale filters (essentially removing the boundary effects present in training correlation filters through spatial constraints). STRCF [26] introduces the spatio-temporal regularization constraints during the filter training process, effectively suppressing boundary effects.

*2.3. Previous Work on Background Information Learning*

These methods [24–26] only model the target and ignore background information, which affects the target tracking performance. To make full use of background information and alleviate boundary effects, Galoogahi et al. [11] proposed the background-aware correlation filter (BACF) algorithm, which extracts HOG features from the target region and dynamically models the foreground and background of the tracking target. The BACF algorithm essentially alleviates boundary effects by expanding the input image block. Although this allows the filter to learn more background information, it also includes too much background noise. In order to suppress boundary effects while expanding the search area and minimize the impact of background noise, Z. Huang et al. [11] designed the learning aberrance repressed correlation filters (ARCFs), which effectively alleviate boundary effects and perform well in various complex tracking scenarios. Mueller et al. proposed a context-aware correlation filter [12], the CACF tracker. The CACF tracker designs four patches close to the object, located in the top, bottom, left, and right directions. The circulant structure of the context patches is the same as that of the object patches. By using these four context patches, the tracker provides more background information, enhancing the tracker's robustness in dealing with complex scenarios.

## 3. Proposed Tracking Methodology

In this chapter, we first introduce the Baseline tracker, and then we will elaborate on our proposed tracker, which includes an aberration suppression module based on spatial attention and an intermittent context learning mechanism. An overview of the tracker is shown in Figure 1. In Table 1, we provide explanations for the variables that will be mentioned later.

As can be seen from Figure 1, our proposed tracker consists of two modules: a spatial attention aberration suppression module and an intermittent context learning mechanism. The spatial attention aberration suppression module effectively suppresses aberrations, while the intermittent context learning mechanism first determines whether the current frame is an aberrance frame or not. If it is an aberrance frame, the APCE criterion is employed to decide whether the global context learning is needed. The APCE criterion is used to determine the necessity of global context learning.

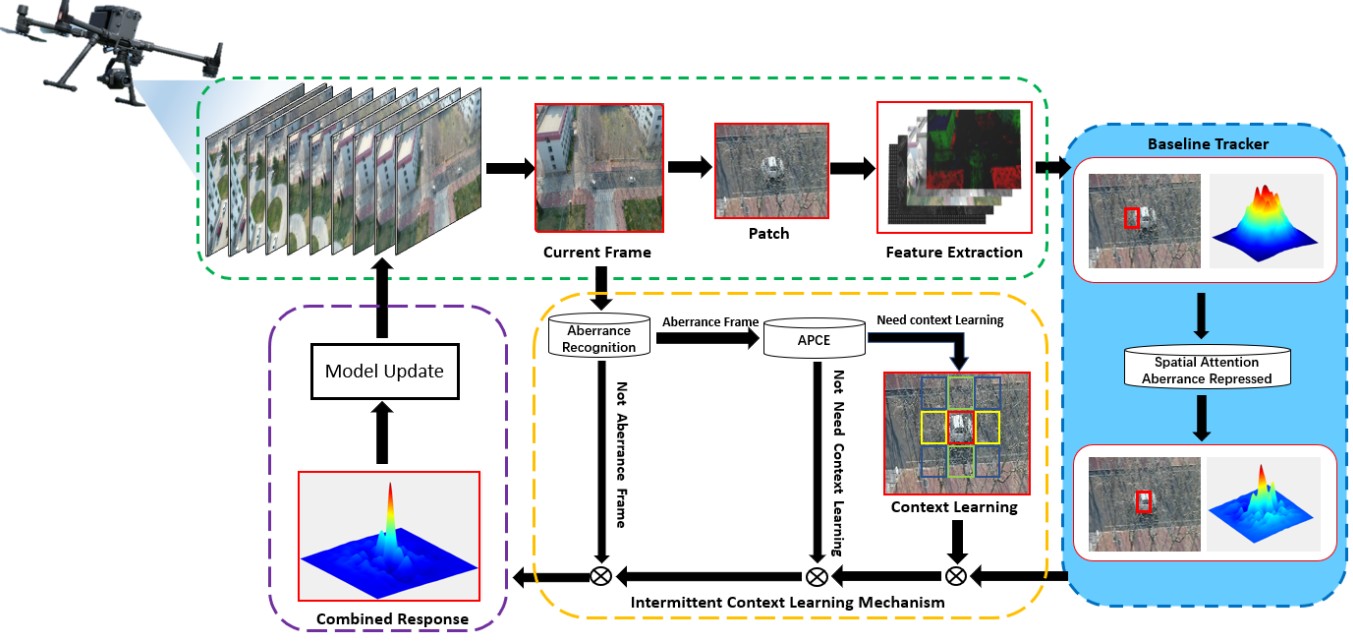

**Figure 1.** The overview of our proposed method.

**Table 1.** Some of the variables that will be used later.

| Denotation | Symbol | Note |
|---|---|---|
| Number of cyclic shifts | N | |
| The size of filter | M | $M \ll N$ |
| Number of feature channels | D | |
| Frame | k | |
| Spatial correlation operator | $\star$ | |
| Crops matrix | P | $P \in R^{M \times N}$ |
| The vectorized ideal response | y | $y \in R^N$ |
| The vectorized samples | $x_k^d$ | $x_k^d \in R^N$ |
| The filter to be learned | w | $w \in R^M$ |
| Penalty coefficient | $\lambda$ | |

### 3.1. Baseline Tracker

In this work, the background-aware correlation filter (BACF) is considered our baseline tracker. The BACF algorithm uses cyclic matrix sampling. In contrast with other correlation filters, it does not perform cyclic shifts around the target but on the entire image. Then, a cropping matrix is used to crop the samples to the desired size. As such, the number of samples is significantly increased, and the negative samples are real rather than artificially stitched together.

BACF aims to minimize the following objective:

$$E(\mathbf{w}_k) = \frac{1}{2} \left\| \mathbf{y} - \sum_{d=1}^{D} \mathbf{P} \mathbf{x}_k^d \star \mathbf{w}_k^d \right\|_2^2 + \frac{\lambda}{2} \sum_{d=1}^{D} \left\| \mathbf{w}_k^d \right\|_2^2 \qquad (1)$$

### 3.2. Aberration Suppression Module

The baseline tracker used in this work can effectively alleviate the boundary effects present in classic correlation filter algorithms. However, when introducing background information, the BACF introduces a large amount of clutter, leading to the misidentification of similar objects around the target. Moreover, in drone scenarios, the environment undergoes drastic changes, easily generating various distortions that are difficult for BACF to handle. The significant spatial shape changes between the two consecutive frames of

response maps can easily lead to target loss. The difference in response maps can reflect the severity of aberration to a certain extent.

In light of this, we first model the aberrations:

$$\|\Delta_k\|_2^2 = \left\|A_{k-1}\left[F_{p,q}\right] - A_k\right\|_2^2 \tag{2}$$

where $A_k$ is the response map of the k-th frame, $p$ and $q$ represent the position differences of the peak values of the response maps in the previous and current frames, and the shift operation $\left[F_{p,q}\right]$ is used to align the peak positions of the two response maps. When aberrance occurs, the similarity of the response maps decreases, resulting in a higher value of $\|\Delta_k\|_2^2$. In order to suppress the generation of aberrance during the filter training process, the training objective of the filter is optimized to minimize the following loss function:

$$E(w_k) = \frac{1}{2}\left\|y - \sum_{d=1}^{D} Px_k^d \star w_k^d\right\|_2^2 + \frac{\lambda}{2}\sum_{d=1}^{D}\left\|w_k^d\right\|_2^2 + \|\Delta_k\|_2^2 \tag{3}$$

However, $\Delta_k$ adopts the same focus on distortion for every point in space. Instead, we should focus on the aberration occurring near the peak of the response map or, in other words, near the tracking target. This is because the aberration in this area has the greatest impact on filter training. The farther away it is from this focal area, the less relevant the aberration becomes, and we should not allocate too much attention to these edge areas. Otherwise, the filter will focus too much on the information in the edge areas, relatively weakening the filter's ability to distinguish the target. In view of this, this paper designs a spatial attention module $Q$ and introduces it into the training stage of the filter. The new objective function is described as follows:

$$E(w_k) = \frac{1}{2}\left\|y - \sum_{d=1}^{D} Px_k^d \star w_k^d\right\|_2^2 + \frac{\lambda}{2}\sum_{d=1}^{D}\left\|w_k^d\right\|_2^2 + \|Q_k\Delta_k\|_2^2 \tag{4}$$

The $Q_k$ attention module in the k-th frame can be obtained from $A_{k-1}$, where the peak coordinates of $A_{k-1}$ are $\left(x_{(k-1),0}, y_{(k-1),0}\right)$. In this paper, the numerical representation of the attention that should be allocated to each area $(x, y)$ in the spatial domain is as follows:

$$Q_k(x, y) = \mu_0 + \alpha e^{-\frac{\left|x - x_{(k-1),0}\right|}{x_{(k-1),0}}} + \alpha e^{-\frac{\left|y - y_{(k-1),0}\right|}{y_{(k-1),0}}} \tag{5}$$

where $\mu_0$ is the base factor and $\alpha$ is the scale factor.

Figure 2 shows the tracking results and response graphs of the proposed tracker in this paper (SAARCF) and the baseline tracker (BACF). The suppression effect of SAARCF on distortion can also be observed from the response map, which is significantly more prominent compared to BACF.

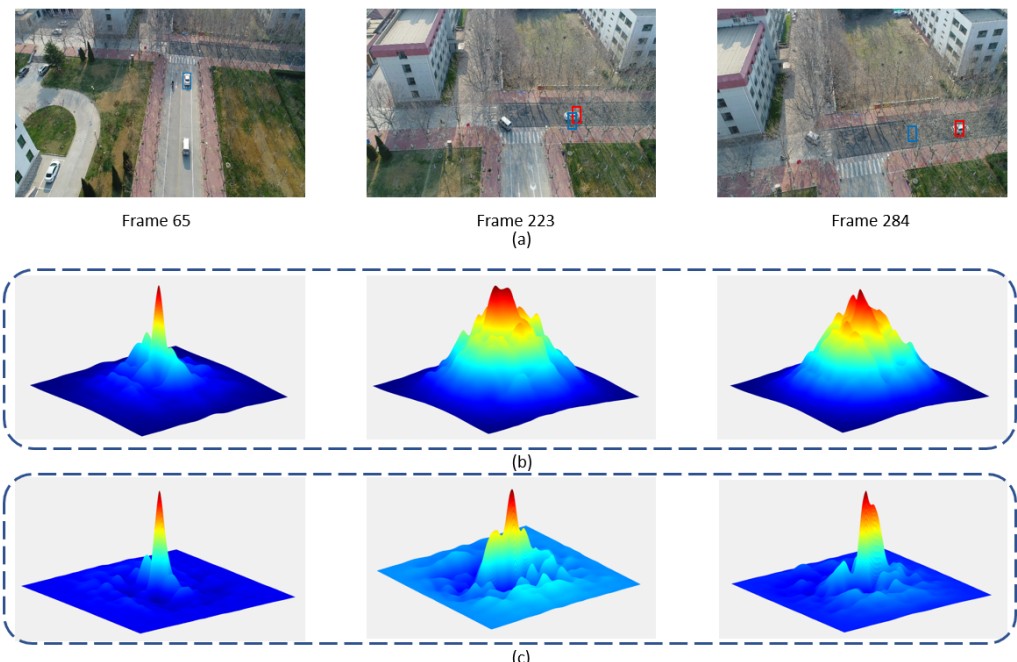

**Figure 2.** Comparison of the baseline tracker and the tracker we propose: (**a**) Comparison of the tracking results of our proposed tracker (red) and the baseline tracker (blue). (**b**) Response map of the baseline tracker (BACF). (**c**) Response map of the tracker proposed herein.

### 3.3. Intermittent Context Learning Mechanism

#### 3.3.1. Aberrance Frame Recognition

In the baseline tracker, only limited context information is included, which makes the tracker prone to drift in situations of rapid movement, occlusion, and background noise. To address this issue, we incorporated global context information into the filter training process. Frame-by-frame context learning is considered highly redundant, as the capture frequency of drone cameras is generally lower than the context change frequency. For example, in a 30 frames per second (FPS) video, the interval between two consecutive time points is 0.03 s, but in reality, the background in an aerial view remains unchanged for a much longer time than 0.03 s [27]. Furthermore, due to the ubiquitous appearance changes in aerial scenes, an unrestricted learning-based single filter is prone to damage.

In light of this, we also need to allocate reasonable attention to consecutive image sequences, focusing on frames with dramatic scene changes (which we refer to as aberrance frames) to fully utilize the limited computing resources. Based on this, we use the $\|\Delta_k\|$ mentioned in Section 3.2 as the criterion for identifying aberrance frames, setting a threshold $G = 2 * \frac{1}{T} \sum_{k=1}^{T} \|\Delta_k\|_2$, where $T$ is the number of image sequences between the current frame ($t$) and the previous aberrance frame ($t-1$). When $\|\Delta_k\|_2 > G$, it indicates that the aberrance is exceptionally noticeable, and the frame is recognized as an aberrance frame ($t$).

#### 3.3.2. Multi-Peak Confidence Level

In regard to the aberrance frames defined in Section 3.3.1, the tracker has a certain level of robustness. In some cases, even without learning the global context, the tracker can rely on its own robustness to counteract the impact of aberrance on tracking results. In this situation, learning the global context during the filter training stage might mistakenly learn some background information, leading to error accumulation and a waste of computational resources. Therefore, after identifying the aberrance frames, it is necessary to first evaluate the confidence of the tracking results at this moment. When the confidence is low, it indicates that the tracker's inherent robustness is no longer sufficient to handle the dramatic scene changes. In such cases, the global context should be learned during the filter training stage.

In this paper, the maximum response value of the filter output and the average peak-to-correlation energy (APCE) [28] are used as metrics to assess the confidence level of the tracking results:

$$\text{APCE} = \frac{|F_{\max} - F_{\min}|^2}{\text{mean}\left(\sum_{x,y}(F_{x,y} - F_{\min})^2\right)} \tag{6}$$

Here, $F_{\max}$ represents the maximum peak of the response map, $F_{\min}$ denotes the minimum peak value, and $F_{x,y}$ indicates the response value at the $(x,y)$ position. When APCE is small, it suggests that the target is significantly affected by surrounding interference, leading to low tracking result confidence. Conversely, a larger APCE implies a higher tracking result confidence. In this study, we establish a threshold L, where L is calculated as the mean APCE of all frames between the current frame and the previous aberrance frame, multiplied by $\theta$ ($\theta$ is a scaling factor). If the current APCE is less than L, we determine that the confidence is low, indicating that the aberrance has a substantial impact on the tracker. Consequently, the tracker is unable to effectively handle the dramatic scene changes, necessitating the learning of the global context.

### 3.3.3. Context Patches Scoring Scheme

We calculate the scores of context patches using Euclidean distances. Context patches have the same dimensions as the target and are located around the target. The score of patch $j$ is specified as follows:

$$Z_j = \frac{\sqrt{w^2 + h^2}}{|OO_j|} z \tag{7}$$

where $|OO_j|$ denotes the Euclidean distance between the target and the context patch $j$ ($j = 1, 2, \ldots, J$), $z$ is a constant, and $w$ and $h$ are the width and height of the target box, respectively.

At this point, for the aberrance frame, the new objective function is described as follows:

$$E(w_k) = \frac{1}{2}\left\|y - \sum_{d=1}^{D} Px_k^d \star w_k^d\right\|_2^2 + \frac{\lambda}{2}\sum_{d=1}^{D}\left\|w_k^d\right\|_2^2 + \|Q_k\Delta_k\|_2^2 + \frac{Z_J}{2}\sum_{j=1}^{J}\left\|\sum_{d=1}^{D} Px_j^d \star w_k^d\right\|_2^2 \tag{8}$$

where the third term is the response of context patches (desired responses are 0) and P is the number of context patches. For non-aberrant frames, the objective function remains as Equation (3).

Next, we need to transform the objective function to the frequency domain. To facilitate computation, we first convert Equation (8) into matrix form as follows:

$$
\begin{aligned}
E(\mathbf{w}_k) = {}& \frac{1}{2}\left\|\mathbf{y} - \mathbf{X}_k\left(\mathbf{I}_D \otimes \mathbf{P}^\top\right)\mathbf{w}_k\right\|_2^2 + \frac{\lambda}{2}\|\mathbf{w}_k\|_2^2 \\
& + \frac{1}{2}\left\|Q_k\left(\mathbf{A}_{k-1}[F_{p,q}] - \mathbf{X}_k\left(\mathbf{I}_D \otimes \mathbf{P}^\top\right)\mathbf{w}_k\right)\right\|_2^2 \\
& + \frac{Z_j}{2}\sum_{j=1}^{J}\left\|\mathbf{X}_j\left(\mathbf{I}_D \otimes \mathbf{P}^\top\right)\mathbf{w}_k\right\|_2^2
\end{aligned} \tag{9}
$$

Now, we will explain the parameters in the objective function for the $k$-th frame. Note that the parameters we discuss below are all with the subscript $k$ omitted. The detailed explanations are as follows.

$\otimes$ is the Kronecker product and $I_D$ is an identity matrix whose size is $D \times D$. F is a kind of mapping: $\hat{q} = \sqrt{N}Fq$, $\mathbf{g} \in C^{DN \times 1}$ and $\mathbf{w} \in R^{DM \times 1}$. This is a more detailed description of these variables: $\mathbf{X}_k$ is the matrix form of the input sample $x$, $\mathbf{X}_P$ is the $p$-th context path. $\mathbf{X} = \left[\text{diag}(\mathbf{x}^1)^\top, \cdots, \text{diag}(\mathbf{x}^D)^\top\right]$, $\mathbf{g} = \left[\mathbf{g}^{1\top}, \cdots, \hat{\mathbf{g}}^{D\top}\right]^\top$ and $\mathbf{X}_p \in C^{N \times DN}$ ($p = 1 \ldots P$).

*3.4. Transformation into Frequency Domain*

The equation above represents the total loss function in matrix form, which is essentially a convolution operation. In order to ensure computational efficiency, it is transformed into the frequency domain as follows:

$$
\begin{aligned}
\hat{E}(w_k, \hat{g}_k) = {} & \frac{1}{2}\|\hat{y} - \hat{X}_k\hat{g}_k\|_2^2 + \frac{\lambda}{2}\|w_k\|_2^2 \\
& + \frac{1}{2}\left\|\hat{Q}_k\big(\hat{A}_{k-1}\big)\big(\hat{A}_{k-1} - \hat{X}_k\hat{g}_k\big)\right\|_2^2 \\
& + \frac{Z_j}{2}\sum_{j=1}^{J}\|\hat{X}_j\hat{g}_k\|_2^2
\end{aligned}
\tag{10}
$$

$$
\text{s.t.} \quad \hat{\mathbf{g}}_k = \sqrt{N}\Big(\mathbf{I}_D \otimes \mathbf{F}\mathbf{P}^\top\Big)\mathbf{w}_k
$$

Here, ˆ represents the discrete Fourier transform (DFT). To further optimize, we introduce a new parameter, which represents the discrete Fourier transform of the $k-1$ frame response signal $A_{k-1}$ after translation. At the $k$-th frame, the response image $A_{k-1}$ of the $k-1$ frame and the value obtained from $A_{k-1}$ can be regarded as constants, which can further simplify the calculation.

*3.5. Optimization through ADMM*

The alternating direction method of multipliers (ADMM) is fast in solving convex optimization problems and has good convergence performance. Equation (10) can be solved using ADMM to obtain the global optimal solution. Therefore, we transform Equation (10) into the augmented Lagrangian multiplier (ALM) form as follows:

$$
\begin{aligned}
\hat{E}(\mathbf{w}_{k\cdots}, \hat{\mathbf{g}}_k, \hat{\zeta}) = {} & \frac{1}{2}\|\hat{\mathbf{y}} - \hat{\mathbf{X}}_k\hat{\mathbf{g}}_k\|_2^2 + \frac{\lambda}{2}\|\mathbf{w}_k\|_2^2 \\
& + \frac{1}{2}\left\|\hat{Q}_k\big(\hat{\mathbf{A}}_{k-1} - \hat{\mathbf{X}}_k\hat{\mathbf{g}}_k\big)\right\|_2^2 + \frac{Z_j}{2}\sum_{j=1}^{J}\|\hat{X}_j\hat{g}_k\|_2^2 \\
& + \hat{\zeta}^\top\Big(\hat{\mathbf{g}}_k - \sqrt{N}\big(\mathbf{I}_D \otimes \mathbf{F}\mathbf{P}^\top\big)\mathbf{w}_k\Big), \\
& + \frac{\mu}{2}\left\|\hat{\mathbf{g}}_k - \sqrt{N}\big(\mathbf{I}_D \otimes \mathbf{F}\mathbf{P}^\top\big)\mathbf{w}_k\right\|_2^2
\end{aligned}
\tag{11}
$$

Here, $\mu$ is the penalty factor and $\hat{\zeta} = \left[\hat{\zeta}^{1\top}, \cdots, \hat{\zeta}^{D\top}\right]^\top$ is an auxiliary variable in the Fourier domain that we introduce with dimensions $DN \times 1$. The advantage of using ADMM is that it can transform the solution of the overall problem into solving two simpler subproblems, that is, the solution of the $k+1$ frame correlation filter can be transformed into solving for $w_{k+1}^*$ and $\hat{g}_{k+1}^*$. The specific solution methods are as follows:

$$
\begin{cases}
\mathbf{w}_{k+1}^* = \arg\min_{\mathbf{w}_k}\Big\{ \dfrac{\lambda}{2}\|\mathbf{w}_k\|_2^2 \\
\qquad\qquad + \hat{\zeta}^\top\Big(\hat{\mathbf{g}}_k - \sqrt{N}\big(\mathbf{I}_D \otimes \mathbf{FP}^\top\big)\mathbf{w}_k\Big) \\
\qquad\qquad + \dfrac{\mu}{2}\Big\|\hat{\mathbf{g}}_k - \sqrt{N}\big(\mathbf{I}_D \otimes \mathbf{FP}^\top\big)\mathbf{w}_k\Big\|_2^2 \Big\} \\[4pt]
\hat{\mathbf{g}}_{k+1}^* = \arg\min_{\mathbf{g}_k}\Big\{ \dfrac{1}{2}\|\hat{\mathbf{y}} - \hat{\mathbf{X}}_k\hat{\mathbf{g}}_k\|_2^2 \\
\qquad\qquad + \dfrac{1}{2}\big\|\hat{Q}_k\big(\hat{\mathbf{A}}_{k-1} - \hat{\mathbf{X}}_k\hat{\mathbf{g}}_k\big)\big\|_2^2 + \dfrac{Z_j}{2}\displaystyle\sum_{j=1}^{J}\|\hat{X}_j\hat{g}_k\|_2^2 \\
\qquad\qquad + \hat{\zeta}^\top\Big(\hat{\mathbf{g}}_k - \sqrt{N}\big(\mathbf{I}_D \otimes \mathbf{FP}^\top\big)\mathbf{w}_k\Big) \\
\qquad\qquad + \dfrac{\mu}{2}\Big\|\hat{\mathbf{g}}_k - \sqrt{N}\big(\mathbf{I}_D \otimes \mathbf{FP}^\top\big)\mathbf{w}_k\Big\|_2^2 \Big\}
\end{cases}
\tag{12}
$$

### 3.5.1. Solution to Subproblem $w_{k+1}^*$

The solution of the subproblem $w_{k+1}^*$ can be easily obtained from the following equation:

$$
\begin{aligned}
\mathbf{w}_{k+1}^* &= (\lambda + \mu N)^{-1}\Big(\sqrt{N}\big(\mathbf{I}_D \otimes \mathbf{PF}^\top\big)\hat{\zeta} + \mu\sqrt{N}\big(\mathbf{I}_D \otimes \mathbf{PF}^\top\big)\hat{\mathbf{g}}_k\Big) \\
&= \Big(\dfrac{\lambda}{N} + \mu\Big)^{-1}(\zeta + \mu\mathbf{g}_k)
\end{aligned}
\tag{13}
$$

The $g_k$ and $\zeta$ can be obtained by Fourier inverse transform:

$$
\begin{cases}
\mathbf{g}_k = \dfrac{1}{\sqrt{N}}\big(\mathbf{I}_D \otimes \mathbf{PF}^\top\big)\hat{\mathbf{g}}_k \\[6pt]
\zeta = \dfrac{1}{\sqrt{N}}\big(\mathbf{I}_D \otimes \mathbf{PF}^\top\big)\hat{\zeta}
\end{cases}
\tag{14}
$$

### 3.5.2. Solution to Subproblem $g_{k+1}^*$

It can be seen that subproblem $g_{k+1}^*$ includes a term $\hat{\mathbf{X}}_k$, which is time-consuming to compute. Therefore, by exploiting the sparsity of $\hat{\mathbf{X}}_k$, in each element of $\hat{y}$ i.e., $\hat{y}(n)$, where $n = 1, 2, \ldots, N$, is solely dependent on:

$$
\begin{aligned}
\hat{\mathbf{g}}_k(n) &= \Big[\operatorname{conj}\big(\hat{\mathbf{g}}_k^1(n)\big), \ldots, \operatorname{conj}\big(\hat{\mathbf{g}}_k^D(n)\big)\Big]^T \\
\hat{\mathbf{x}}_k(n) &= \Big[\hat{\mathbf{x}}_k^1(n), \hat{\mathbf{x}}_k^2(n), \ldots, \hat{\mathbf{x}}^D(n)\Big]^T
\end{aligned}
\tag{15}
$$

where operator conj(. ) denotes the complex conjugate operation.

From this, subproblem $g_{k+1}^*$ is decomposed into $N$ smaller problems:

$$
\begin{aligned}
\hat{\mathbf{g}}_{k+1}(n)^* = \arg\min_{\mathbf{g}_k(n)}\Big\{ &\dfrac{1}{2}\Big\|\hat{\mathbf{y}}(n) - \hat{\mathbf{x}}_k^\top(n)\hat{\mathbf{g}}_k(n)\Big\|_2^2 \\
&+ \dfrac{1}{2}\Big\|\hat{Q}_k\big(\hat{\mathbf{A}}_{k-1} - \hat{\mathbf{x}}_k^\top(n)\hat{\mathbf{g}}_k(n)\big)\Big\|_2^2 + \dfrac{Z_j}{2}\displaystyle\sum_{j=1}^{J}\Big\|\hat{x}_j^T(n)\hat{g}_k(n)\Big\|_2^2 \\
&+ \hat{\zeta}^\top\big(\hat{\mathbf{g}}_k(n) - \hat{\mathbf{w}}_k(n)\big) \\
&+ \dfrac{\mu}{2}\|\hat{\mathbf{g}}_k(n) - \hat{\mathbf{w}}_k(n)\|_2^2 \Big\}
\end{aligned}
\tag{16}
$$

The solution for each smaller problem after decomposition can be easily obtained:

$$
\hat{\mathbf{g}}_{k+1}(n)^* = (I_N + Q^2)^{-1} \left( \hat{\mathbf{x}}_k(n)\hat{\mathbf{x}}_k^\top(n) + \sum_{j=1}^{J} Z_j^2 \hat{\mathbf{x}}_j(n)\hat{\mathbf{x}}_j^\top(n) + \mu(I_N + Q^2)^{-1} \right)^{-1}
$$
$$
\left( \hat{\mathbf{x}}_k(n)\hat{\mathbf{y}}(n) + Q^2 \hat{\mathbf{x}}_k(n)\hat{\mathbf{A}}_{k-1} - \hat{\boldsymbol{\zeta}}(n) + \mu \hat{\mathbf{w}}_k(n) \right)
$$

(17)

### 3.5.3. Update of Lagrangian Parameter

The Lagrangian parameters are updated by the following equation:

$$
\hat{\boldsymbol{\zeta}}_{k+1}^{(i+1)} = \hat{\boldsymbol{\zeta}}_{k+1}^{i} + \mu \left( \hat{\mathbf{g}}_{k+1}^{*(i+1)} - \hat{\mathbf{w}}_{k+1}^{*(i+1)} \right)
$$

(18)

where $i$ denotes the $i$-th iteration and $\hat{\mathbf{w}}_{k+1}^{*(i+1)} = \left( \mathbf{I}_D \otimes \mathbf{FP}^\top \right) \mathbf{w}_{k+1}^{*(i+1)}$

### 3.5.4. Update of Appearance Model

The appearance model $\hat{\mathbf{x}}^{\mathrm{model}}$ is updated as follows:

$$
\hat{\mathbf{x}}_k^{\mathrm{model}} = (1 - \eta)\hat{\mathbf{x}}_{k-1}^{\mathrm{model}} + \eta \hat{\mathbf{x}}_k
$$

(19)

Here, $\eta$ is the learning rate of the appearance model.

## 4. Experiment

In this chapter, to evaluate the performance of the proposed algorithm, we conduct quantitative and qualitative experiments on two challenging UAV target tracking datasets, UAVDT and Visdrone2018. The results are compared with 11 other state-of-the-art trackers that use handcrafted features, including: AutoTrack, ECO-HC, ARCF (HOG + CN), ARCF-H (HOG), SRDCF, BACF, Staple, DSST, SAMF, KCF, and CSK. To ensure the fairness of the real-time evaluation, the code obtained from each author's webpage was uniformly run on the author's computer, rather than directly using their evaluation results. If readers are interested in understanding the specific parameters set by the authors during their testing, they can refer to the literature [11,21,23,24,29–34] for detailed information. The tracker's settings and parameters used in my testing are the same as those used by the authors in the original study. The experiments can be divided into overall performance evaluation and sub-attribute performance evaluation. These datasets cover various complex scenes and diverse attributes in the UAV tracking domain and are widely used to verify the effectiveness of tracking algorithms in UAV scenarios.

### 4.1. Experimental Platform and Parameters

Regarding the aberration suppression module mentioned in Section 3.2, we set its parameters as $\mu_0 = 2$ and $\alpha = 1.5$ during the testing. For the intermittent learning context mechanism mentioned in Section 3.3, we set $z = 0.24$ and $\theta = 0.5$ during the testing. The number of iterations for ADMM is set to 5, and the learning rate is 0.0192. All 12 trackers are run on a computer equipped with an Intel(R) Core(TM) i5-10300H CPU @ 2.50 GHz processor, and 16 GB RAM using MATLAB R2019a.

### 4.2. Evaluation Methodology

We use the one-pass evaluation (OPE) method, which only provides the ground truth for the target's first frame during the entire testing process and does not initialize it subsequently. We adopt an efficient set of evaluation rules proposed by Wu et al. [35], which reasonably assess the performance of trackers from the aspects of the distance precision rate (DPR) and overlap success rate (OSR). The center location error (CLE) refers to the Euclidean distance between the center point of the bounding box and the tracking result [36]. The DPR for a single sequence represents the proportion of image sequences with a CLE less than a certain threshold, typically using a 20-pixel CLE for sorting trackers.

Similarly, the "intersection over union (IoU)" can also be used for sorting. The numerator of the IoU is the area of the intersection between the test result box and the ground truth, while the denominator is the area of the union. The OSR for a single sequence represents the proportion of image sequences with an IoU greater than a certain threshold.

### 4.3. Benchmarks for Experiments

The UAVDT benchmark consists of 100 video sequences, recorded at 30 frames per second (fps) with a resolution of 1080 × 540 pixels [37]. UAVDT can be used for single-object detection and single-object tracking, and the single-object tracking part includes 50 video sequences with 9 attributes: camera motion (CM), illumination variation (IV), long-term tracking (LT), large occlusion (LO), object motion (OM), object blur (OB), background clutter (BC), small object (SO), and scale variation (SV).

The Visdrone benchmark is captured by cameras mounted on various drones, covering a wide range of aspects, including location (from 14 different cities in China, thousands of kilometers apart), environment (urban and rural), objects (pedestrians, vehicles, bicycles, etc.), and density (sparse and crowded scenes). The dataset is collected using different drone platforms (i.e., different drone models) in different scenes, as well as different weather and lighting conditions. The single-object tracking part of Visdrone contains 50 video sequences, including 13 attributes: aspect ratio change (ARC), background clutter (BC), camera motion (CM), fast motion (FM), full occlusion (FO), illumination variation (IV), low resolution (LR), out-of-view (OV), partial occlusion (PO), similar occlusion (SO), similar object (SO), scale variation (SV), and viewpoint change (VC).

### 4.4. Overall Performance Evaluation

The overall performance evaluation results of the proposed SAARCF algorithm on the UAVDT benchmark are shown in Figure 3. It should be noted that the performance evaluation values in the figure are percentages. The precision and success rate are 0.743 and 0.514, respectively, which represent improvements of 12.1% and 12.7% compared to the baseline tracker. Moreover, compared to other advanced trackers, it generally has a better overall performance.

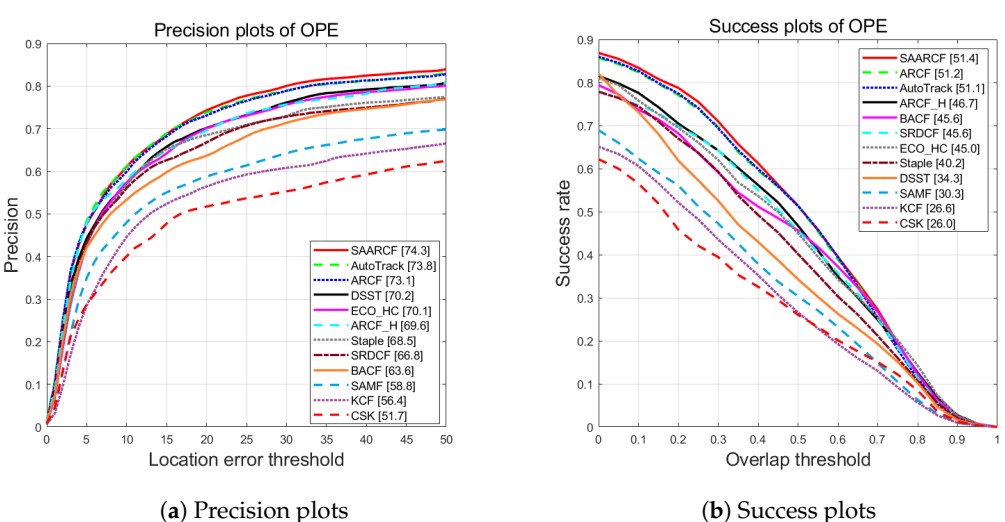

(**a**) Precision plots          (**b**) Success plots

**Figure 3.** Evaluation of tracker attributes on the Visdrone dataset.

The overall performance evaluation results of the SAARCF algorithm on the Visdrone benchmark are shown in Figure 4, and it should be noted that the performance evaluation values in the figure are all percentages. The precision and success rate are 0.798 and 0.719, respectively, which are 2.6/2.4% higher than those of the baseline tracker, and it also has a better overall performance compared to other advanced trackers. Although the precision is

slightly lower than ECO-HC by 0.01, the success rate is higher than ECO-HC by 0.19, so the comprehensive performance is still better than ECO-HC.

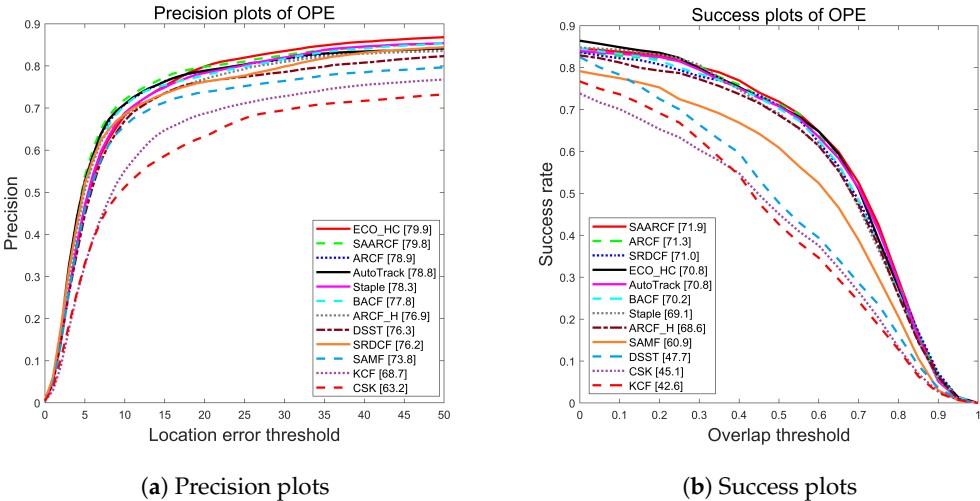

(**a**) Precision plots        (**b**) Success plots

**Figure 4.** Evaluation of tracker attributes on the Visdrone dataset.

### 4.5. Attribute Evaluation

Figures 5 and 6 demonstrate the comparison of precision and success rates for 8 attributes on the UAVDT dataset. As can be seen from the figures, SAARCF achieves an improvement of more than 10% in all 8 attributes compared to the baseline tracker. In contrast to the other trackers, SAARCF achieves scores of SV (0.647), CM (0.718), BC (0.668), OM (0.672), and LO (0.548), while the overall performance of Autotrack, which is the second-best, improves by SV (1.7%), CM (3.2%), BC (3.2%), OM (1.3%), and LO (4.9%). SAARCF obtains the highest success rates for the attributes SV (0.457), CM (0.493), BC (0.441), and SO (0.558), with improvements over Autotrack of SV (0.4%), CM (2.2%), BC (1.1%), OM (1.3%), and OB (1.6%). Figure 7 provides a more intuitive comparison of the performance of various trackers in the 8 attributes.

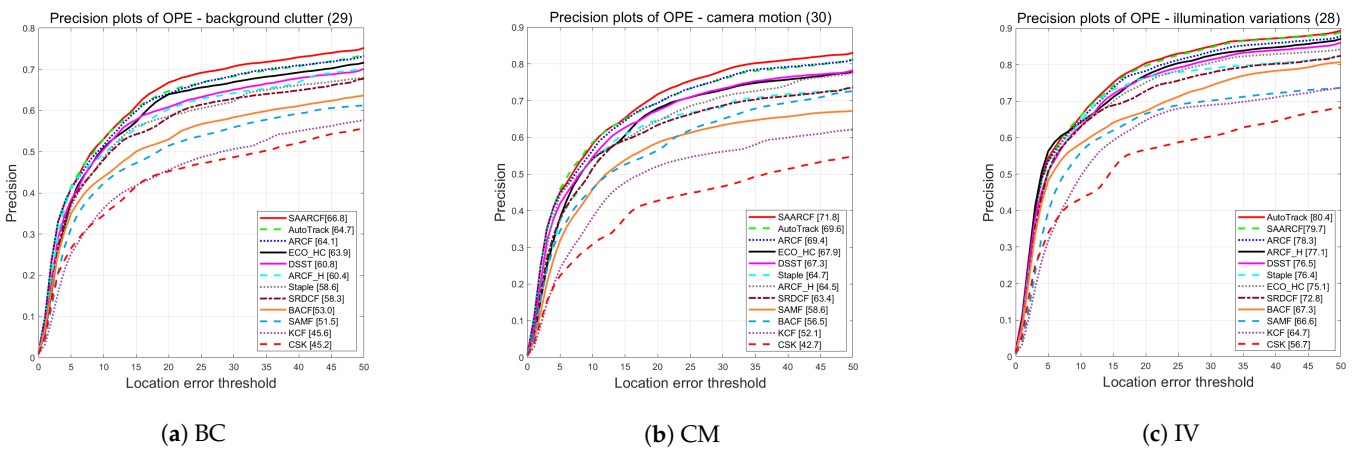

(**a**) BC        (**b**) CM        (**c**) IV

**Figure 5.** *Cont.*

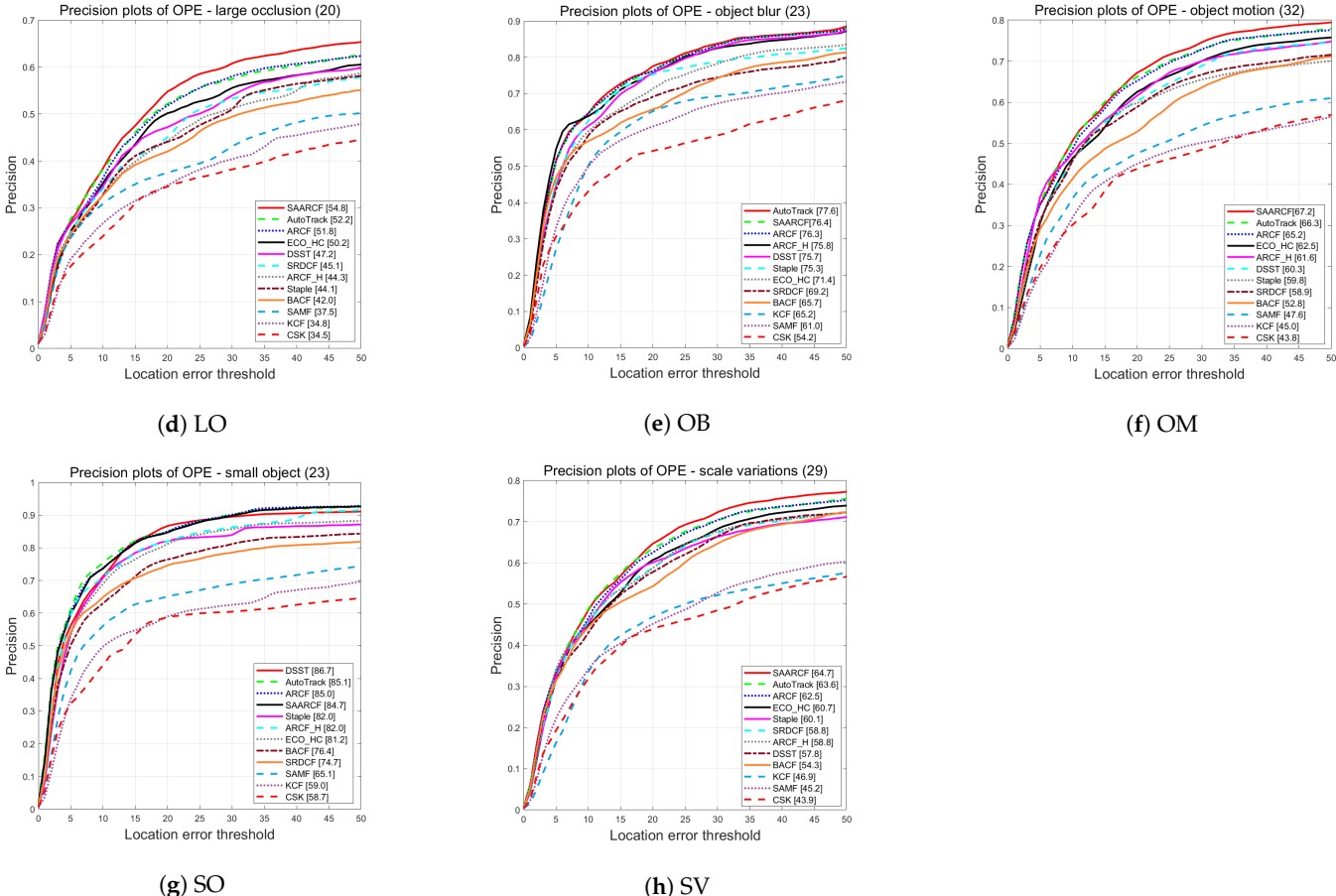

**Figure 5.** Tracker's precision in attribute testing on the UAVDT dataset.

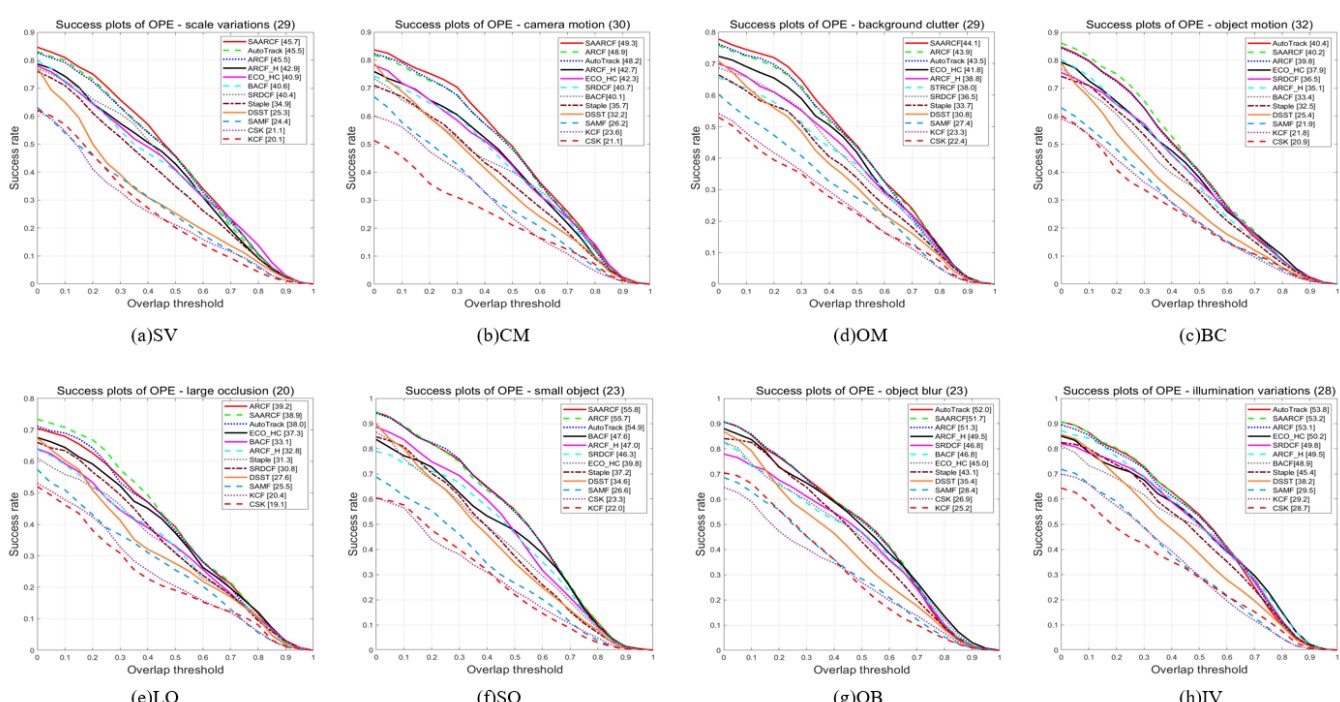

**Figure 6.** Tracker'ssuccess rate in attribute testing on the UAVDT dataset.

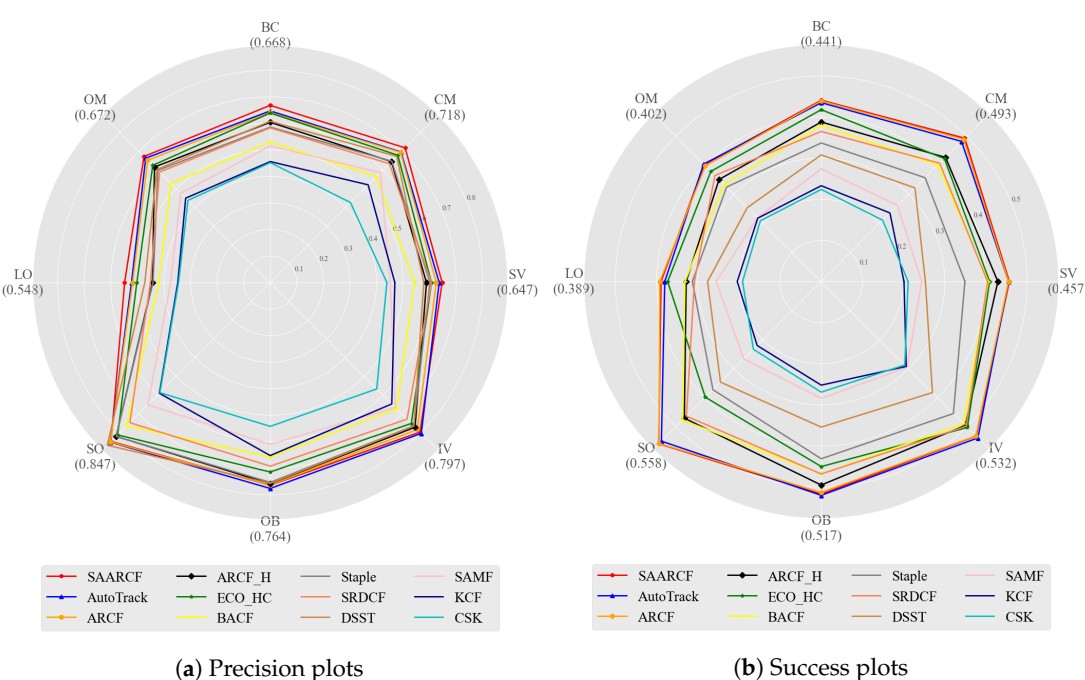

(**a**) Precision plots  (**b**) Success plots

**Figure 7.** Evaluation of tracker attributes on the UAVDT dataset.

The evaluation results for the 13 attributes of the Visdrone dataset are shown in Figure 8. Figure 8 presents the comparison of precision and success rates for trackers on the 13 attributes in the Visdrone dataset. SAARCF outperforms the baseline tracker in all 13 attributes. The precision rates were: VC (0.819), SV (0.766), SO (0.641), PO (0.737), OV (0.802), LR (0.706), IV (0.856), FO (0.701), FM (0.746), CM (0.799), BC (0.663), and ARC (0.799). The success rates were: VC (0.787), SV (0.674), SO (0.540), PO (0.698), OV (0.693), LR (0.461), IV (0.818), FO (0.664), FM (0.669), CM (0.722), BC (0.543), and ARC (0.661). SAARCF ranks first in both the success and precision rates for the CM, FM, and SO attributes. Figures 9–11 show the precision and success rates of various trackers for these three attributes, respectively.

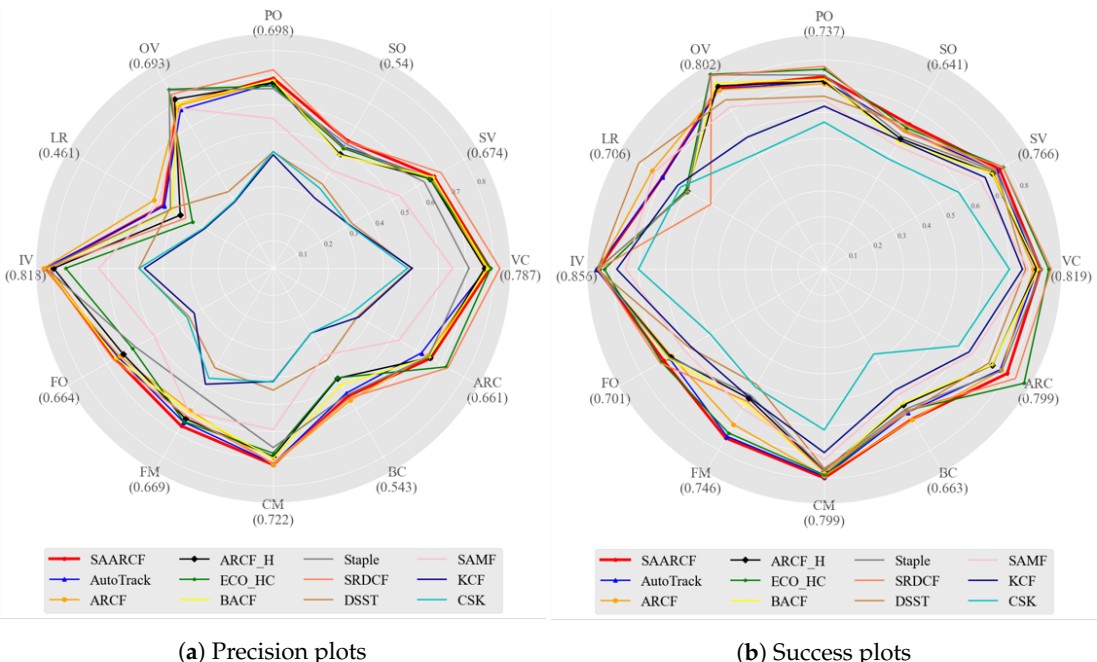

(**a**) Precision plots  (**b**) Success plots

**Figure 8.** Evaluation of tracker attributes on the Visdrone dataset.

Due to the introduction of the spatial attention aberrance suppression module, SAARCF can effectively cope with the distortion of the environment and targets. This is evidenced by the highest scores achieved by SAARCF in the CM and MO attributes in Figures 9 and 10. Moreover, Figure 11 shows that SAARCF performs better in dealing with similar target interference compared to other trackers, which further confirms that learning the global context can effectively enhance the filter's ability to distinguish targets.

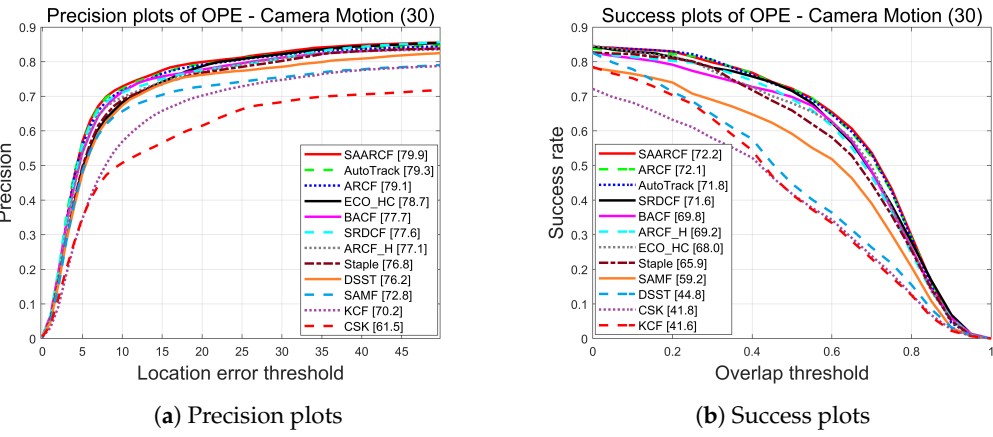

(**a**) Precision plots                    (**b**) Success plots

**Figure 9.** Comparison of CM properties on the Visdrone dataset.

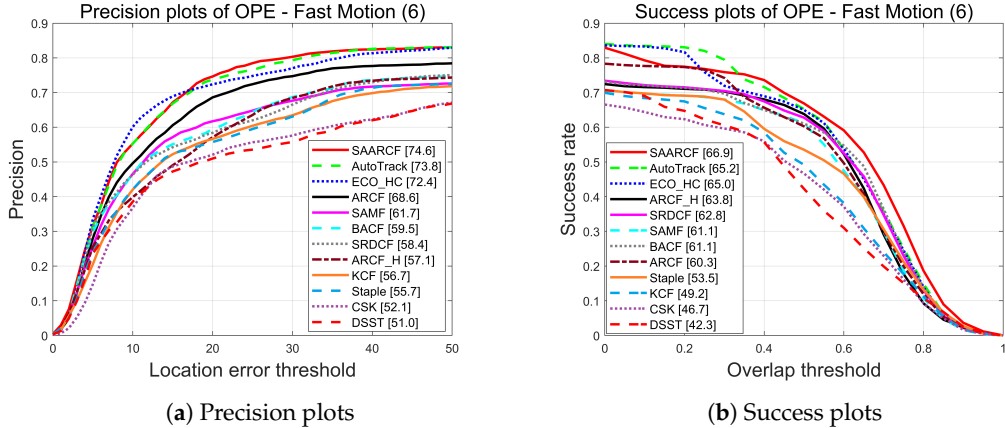

(**a**) Precision plots                    (**b**) Success plots

**Figure 10.** Comparison of FM properties on the Visdrone dataset.

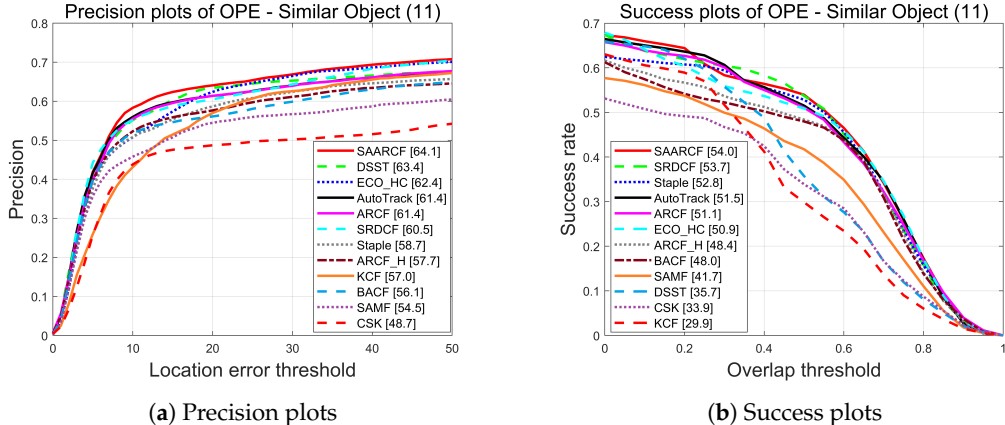

(**a**) Precision plots                    (**b**) Success plots

**Figure 11.** Comparison of SO properties on the Visdrone dataset.

*4.6. Real-Time Evaluation*

We conducted a real-time evaluation of SAARCF and 11 other trackers on the UAVDT and Visdrone datasets, with the results displayed in Table 2. As shown in Table 2, our proposed SARRCF algorithm can meet the requirements of real-time tracking. To verify the effectiveness of our proposed intermittent learning context mechanism (ILCM), we compared the FPS of three scenarios: the baseline tracker with the aberration suppression module (baseline + SAAR), baseline tracker with aberration suppression module and the frame-by-frame learning of global context (Baseline + SAAR + LCFF), and baseline tracker with aberration suppression module and ILCM (baseline + SAAR + ILCM). The results are shown in Table 3. As can be seen from Table 3, our proposed ILCM can effectively save computational resources and improve the computation speed.

**Table 2.** FPS on UAVDT and Visdrone.

| Tracker | Benchmarks | UAVDT | Visdrone |
|---|---|---|---|
| SAARCF | | 28.54 | 24.47 |
| AutoTrack | | 38.78 | 48.84 |
| ARCF | | 31.44 | 26.89 |
| ECO-HC | | 73.55 | 59.96 |
| BACF | | 39.96 | 42.24 |
| SRDCF | | 15.68 | 11.24 |
| ARCF-H | | 42.33 | 47.97 |
| Staple | | 68.84 | 92.74 |
| DSST | | 130.24 | 101.44 |
| SAMF | | 17.22 | 7.94 |
| KCF | | 1283.77 | 498.66 |
| CSK | | 1792.88 | 624.33 |

**Table 3.** FPS in three scenarios.

| Modules | Benchmarks | UAVDT | Visdrone |
|---|---|---|---|
| Baseline + SAAR | | 32.57 | 27.29 |
| Baseline + SAAR + LCFF | | 22.37 | 19.74 |
| Baseline + SAAR + ILCM | | 28.54 | 24.77 |

*4.7. Qualitative Experimental Analysis*

To more intuitively analyze the effectiveness of SAARCF, we selected five video sequences from the UAVDT dataset for evaluation. We compared SAARCF with two other top trackers (AutoTrack and ECO-HC) and a baseline tracker (BACF). The tracking results are shown in Figure 12.

In Seq1, all four trackers can stably track the target before any occlusion occurs. At frame 153, leaves partially occlude the target, but the four trackers still do not exhibit tracking drift. However, at frame 214, the vehicle is mostly occluded, and only SAARCF still maintains the target. This indicates that, during occlusion, SAARCF learns the global context of the target and incorporates more background information. Therefore, even if most of the target area is occluded and the available target information is limited, the target can still be accurately recognized. However, at frame 279, the target is completely occluded, and even SAARCF cannot track it.

In Seq2, although the illumination changes significantly, all four trackers can still accurately track the target. At frame 150, a car passes near the target, and the baseline tracker fails to distinguish the object similar to the target, tracking the wrong target instead. At frame 356, the camera suddenly rotates rapidly, causing ECO-HC to experience tracking drift, while AutoTrack and SAARCF can still accurately follow the target.

In Seq3, as the camera moves upward, the target is completely occluded. Between frames 30 and 36, the target is fully occluded, but when it reappears at frame 37, it is imme-

diately captured by SAARCF. This shows that SAARCF has a certain level of robustness to complete occlusion.

In Seq4 and Seq5, there is some disturbance around the target, accompanied by sudden camera movements and partial occlusion. It can be seen that only SAARCF performs well in completing the tracking task. Especially at frame 429 in Seq5, when the target suddenly jumps to shoot, only SAARCF successfully follows the target. This indicates that SAARCF has a better robustness to sudden changes in the environment or target.

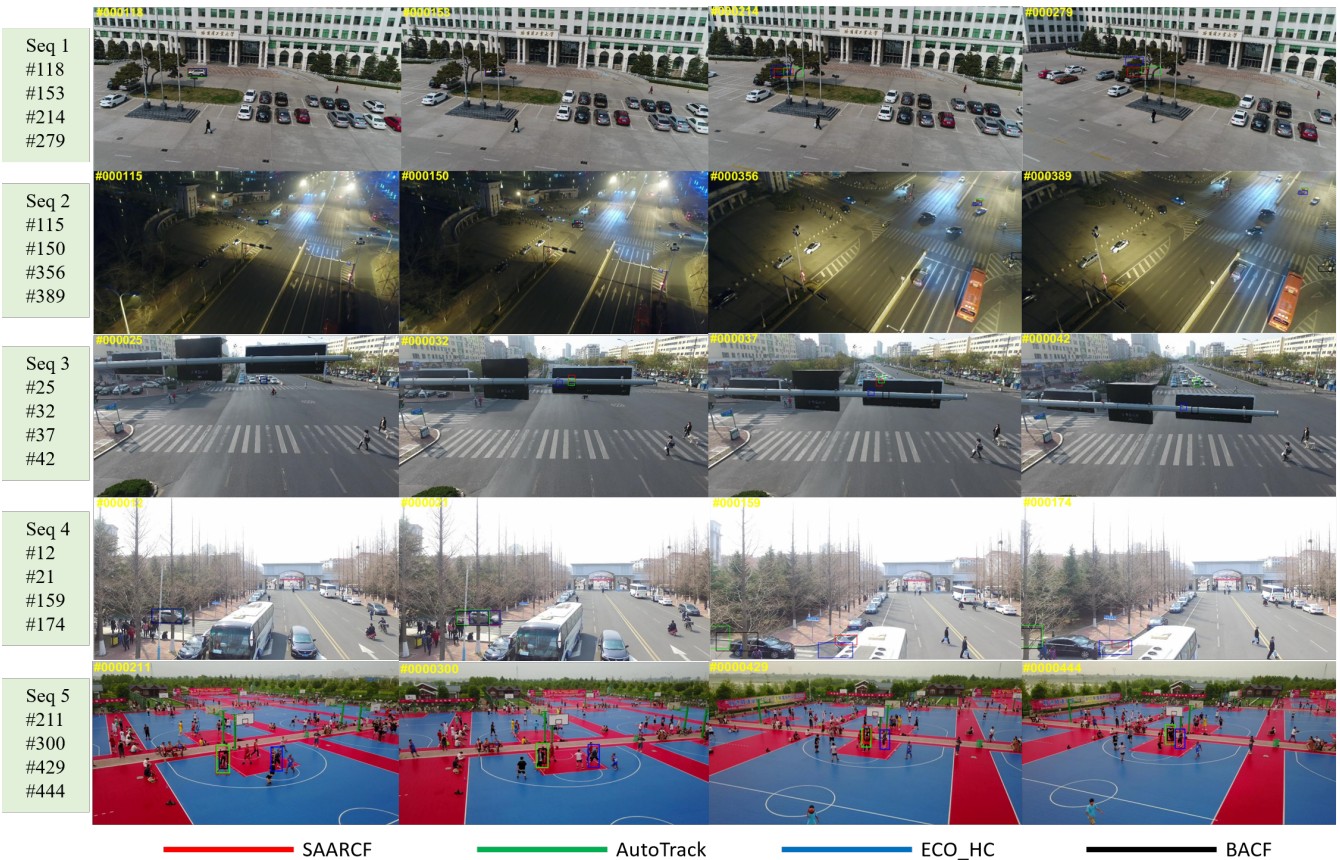

**Figure 12.** The visualization of tracking results.

## 5. Conclusions

To address the drastic changes in scenes and targets from the perspective of UAVs, this paper proposes SAARCF, and in order to fully utilize background information under limited computational resources, an intermittent context learning mechanism is also introduced. Extensive tests of the proposed algorithm were conducted on the UAVDT and Visdrone datasets, and the experimental results show that the proposed SAARCF outperforms the state-of-the-art trackers in overall performance, with a particularly outstanding performance in attributes such as camera movement, rapid target movement, and partial occlusion.

At the same time, there are some shortcomings in this work, namely the fact that it does not take into account the need for target scale estimation in complex environments when the target size changes, in order to adjust the size of the target bounding box and avoid learning too much background information due to an excessively large bounding box. The next step in this research will focus on implementing target scale adaptation.

**Author Contributions:** Investigation, J.H. and L.Y.; writing—original draft preparation, Z.Z.; writing—review and editing, Y.H. and X.L.; supervision, H.G. All authors have read and agreed to the published version of the manuscript.

**Funding:** This research was supported by the National University of Defense Technology Research Grant in 2023.

**Conflicts of Interest:** The authors declare no conflict of interest.

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
