# Peer review of "Towards Robust Visual Tracking for Unmanned Aerial Vehicle with Spatial Attention Aberration Repressed Correlation Filters"

_drones, doi:10.3390/drones7060401_

Round 1

Reviewer 1 Report

Dear authors,

The manuscript titled “Towards Robust Visual Tracking for Unmanned Aerial Vehicle with Spatial Attention Aberration Repressed Correlation Filters” proposed the computing framework SAARCF, which can better cope with UAV target tracking challenges. The experiment justified the better performance than state-of-the-art trackers. The process of the research is of high quality. However, some trivial grammatical points listed below could to be modified:

1.     In lines 7-8, why inconsistent capitalized initials in spatial attention aberration Repressed correlation filter (SAARCF) with only one word emphasized? Rephrasing them with italics or all with capitalized initials would be advisable.

2.     In line 86, ‘The global context’ --> ’ the global context’.

3.     In line 119, is the name of the author 'Bolme'?

4.     In lines 332-334, I suspect misuse the word ‘instead’ although the meaning could be grasped by the context clues.

I recommend to polish the English more intensively. Additionally, if possible, the accessibility of the data and codes of the algorithms could be interesting for critical readers.

--Comments Over—

Dear authors,

The manuscript titled “Towards Robust Visual Tracking for Unmanned Aerial Vehicle with Spatial Attention Aberration Repressed Correlation Filters” proposed the computing framework SAARCF, which can better cope with UAV target tracking challenges. The experiment justified the better performance than state-of-the-art trackers. The process of the research is of high quality. However, some trivial grammatical points listed below could to be modified:

1.     In lines 7-8, why inconsistent capitalized initials in spatial attention aberration Repressed correlation filter (SAARCF) with only one word emphasized? Rephrasing them with italics or all with capitalized initials would be advisable.

2.     In line 86, ‘The global context’ --> ’ the global context’.

3.     In line 119, is the name of the author 'Bolme'?

4.     In lines 332-334, I suspect misuse the word ‘instead’ although the meaning could be grasped by the context clues.

--Comments Over—

Author Response

Dear reviewer:

thank you very much for reviewing our manuscript and providing valuable feedback and suggestions. Under your guidance, we have made corrections to the grammar errors in the text and have taken into account your suggestions. The following are the modifications we made regarding the grammar errors you pointed out:

  1. Regarding your first suggestion, we have capitalized the initials of "Spatial Attention Aberration Repressed Correlation Filter."

  1. In line 86, "The global context" has been modified to "The global context."

Regarding your third suggestion, I have carefully reviewed the corresponding literature and confirmed that the author is indeed Bolme.

According to your fourth suggestion, I realized that I did indeed misuse "instead," so I have removed it.

Thank you for bringing these to my attention. I have thoroughly checked the English grammar of the manuscript. Once again, I appreciate your time and effort in reviewing my article.

Reviewer 2 Report

The article is well-detailed in each of the stages presented, which will surely be appreciated by those who wish to reproduce the experience. Likewise, the detail of the results with the comparison with several other known models adds value. It is a long article, but complete, so in my judgment, it has the right balance. 

Author Response

Dear reviewer:

We are very grateful for your kind appraisal. Thank you for your time and dedication in reviewing our manuscript. We are honored to have had the opportunity to receive feedback from someone with your level of expertise.

Reviewer 3 Report

The study proposes a high-performing tracking method for UAV videos. Overall, the manuscript is well-structured, however minor issues regarding the study must be addressed to make the research ready for publication. 

Firstly, it is mentioned in Experiments section that all the 11 trackers used for comparison have been re-applied by the authors for the sake of fairness. In my opinion, re-running of codes by the authors may have resulted in non-optimal performance evaluation. Actual fairness can only be judged if the evaluation of all available trackers including the proposed tracker, has been conducted by a 3rd Party. Therefore, the authors should at least mention the (optimal) performance of these trackers reported by their respective authors, in similar testing scenario. The authors can add a table may be, highlighting the reported performance (that would be of course at optimal settings and preferences used by actual authors).

Secondly, majority of the references added in the manuscript are related to conference proceedings. Please add more related references from high ranked journal publications to further authenticate the proposed methodology.

Moreover, proof-reading of manuscript is also required.  Figure 1 seems to be placed inappropriately. There is actually no need to show the results in Introduction. Additionally, it is also suggested to remove abbreviations from the abstract.

Overall, the quality of the manuscript is exceptional. However, please address the comments as suggested.

Author Response

Dear reviewer:

thank you very much for reviewing our manuscript and providing valuable feedback and suggestions. We are honored to have had the opportunity to receive feedback from someone with your level of expertise.

Regarding your first suggestion, I apologize for any misunderstanding caused by my unclear expression. I obtained the source code for these 11 trackers from the authors' respective webpages, and when running these codes, I strictly maintained the settings and parameters as described in the original papers. Therefore, I actually compared my best performance with the best performance of these trackers as reported in the literature. My test results are consistent with the authors' original test results.

The reason for running these codes on my computer was that the same tracker can have different FPS (frames per second) when run on different performance computers. So, in order to ensure fairness in the real-time evaluation, I needed to run these 11 trackers on the same machine.

Furthermore, based on your guidance, I have added references in the manuscript to allow readers to directly verify the experimental details.

In order to avoid further misunderstandings, I have made modifications to my original statement in line 335 and added references. The specific modifications are as follows:

“To ensure the fairness of real-time evaluation, I will run the code obtained from each author's webpage on my computer uniformly, rather than directly using their evaluation results.If readers are interested in understanding the specific parameters set by the authors during their testing, they can refer to the literature[31-40] for detailed information. The tracker's settings and parameters used in my testing are the same as those used by the authors in the original study”

Regarding your second suggestion, I have extensively read high-level journals published by IEEE and cited relevant articles in the introduction and related works sections. Additionally, while some of the references are conference papers, they are from prestigious conferences such as CVPR, ICCV, and IRCA, which are considered top-tier conferences in the field of computer vision. I believe that these references provide valuable support for validating the proposed method.

Regarding your third suggestion, thank you for bringing it to my attention. I realize that Figure 1 should not have been included in the introduction, and I have removed it from that section.

Regarding your fourth suggestion, I have removed the abbreviations from the abstract.

Thank you again for your valuable feedback, and we appreciate your time and effort in reviewing our manuscript.

Reviewer 4 Report

1. In Equation 5 all symbols are not defined, what are α and μ0. Please check the full paper, all symbols should be defined.

2. Line 279, "Where", w should be a small letter. "," should not use here.

3. Authors should care about typing and grammatical mistakes.

4. Please add some more recent papers to the related works. Some important and related works are like as below: 10.3390/drones7020089, 10.1109/TIV.2022.3163315

Minor checks are required to increase readability.

Author Response

Dear reviewer:

Thank you very much for reviewing our manuscript and providing valuable feedback and suggestions. We are honored to have had the opportunity to receive feedback from someone with your level of expertise.

Regarding your suggestions, I have made the following modifications:

In response to your first suggestion: I have defined α and μ0 in line 225.

In response to your second suggestion: I have changed "W" to lowercase and removed the comma from that location.

In response to your third suggestion: I have personally reviewed the manuscript's grammar and made some modifications to address any inappropriate expressions.

In response to your fourth suggestion: I have carefully read the papers you recommended and cited them in the introduction and related works sections.

Once again, I appreciate your time and effort in reviewing our manuscript.

Thank you.

Round 2

Reviewer 4 Report

Thanks for your effort. The authors address all of my concerns successfully. The paper could be accepted.